# Impact of Cassava Cultivars on Stylet Penetration Behavior and Settling of *Bemisia tabaci* Gennadius (Hemiptera: Aleyrodidae)

**DOI:** 10.3390/plants13223218

**Published:** 2024-11-15

**Authors:** Sudarat Pimkornburee, Supawadee Pombud, Kumri Buensanteai, Weravart Namanusart, Sukanya Aiamla-or, Jariya Roddee

**Affiliations:** 1School of Crop Production Technology, Institute of Agricultural Technology, Suranaree University of Technology, Nakhon Ratchasima 30000, Thailand; m6500016@g.sut.ac.th (S.P.); supawadeepb@gmail.com (S.P.); kumrai@sut.ac.th (K.B.); sukanya.aia@sut.ac.th (S.A.-o.); 2Department of Plant Science, Faculty of Agricultural Innovation and Technology, Raja Mangala University of Technology Isan, Nakhon Ratchasima 30000, Thailand; weravart.na@rmuti.ac.th

**Keywords:** *Bemisia tabaci*, cassava mosaic virus, electrical penetration graph, host preference, trichome

## Abstract

This study investigates the settling preferences and feeding behavior of the *Bemisia tabaci* whitefly on six cassava cultivars using electrical penetration graph techniques. Six distinct electrical penetration graph waveforms—non-probing, stylet pathway, phloem salivation, phloem ingestion, intracellular puncture, and xylem feeding—were identified and analyzed. Significant differences in the frequency and duration of these waveforms were observed among the cassava cultivars. The whiteflies spent the majority of their time in the non-probing phase, particularly on the Huaybong 80, Kasetsart 50, Rayong 9, and Rayong 72 cultivars. CMR-89 cultivar exhibited higher total probe durations in the phloem salivation and ingestion waveforms, suggesting a greater potential for transmission of the Sri Lankan cassava mosaic virus. The study also examined trichome density and size across the cassava cultivars, revealing that CMR-89 had the highest density and small trichomes, while Huaybong 80 had the lowest density. Trichome characteristics significantly impacted whitefly behavior: larger trichomes were negatively correlated with whitefly settling, whereas higher trichome density was positively correlated with longer settling durations. These findings indicate that trichome-based resistance mechanisms are crucial in whitefly deterrence. Overall, the results suggest that cultivars with lower trichome density and larger trichomes are more resistant to whitefly infestation and subsequent Sri Lankan cassava mosaic virus transmission. These insights are valuable for cassava breeding programs focused on enhancing pest resistance, highlighting the importance of trichome characteristics in developing more resilient cassava varieties.

## 1. Introduction

Cassava (*Manihot esculenta* Crantz, family Euphorbiaceae) is an important root crop in Africa, Asia, and South America. It provides a livelihood to over 500 million people and is among the most important food staples worldwide [1]. Cassava is the world’s third-largest source of carbohydrates for human food and raw material for starch-based industries. However, its cultivation faces significant challenges from various pests and diseases, including over 20 viral diseases associated with vegetative propagation. Another factor increasingly affecting its production is cassava mosaic disease (CMD), resulting in losses estimated at USD 1.6 billion [2,3,4]. CMD, caused by cassava mosaic geminiviruses (family Geminiviridae: genus Begomovirus), is a major threat to global agriculture. Cassava mosaic geminiviruses are one of the top 10 viruses that affect economically important crops, significantly influencing cassava production in Africa and the Indian subcontinent [3,5,6,7]. The devastating impact of CMD was first observed in East Africa in 1894, and the disease has since spread across Africa, India, and Sri Lanka, largely attributed to geminiviruses and their vector, *Bemisia tabaci* [8,9,10]. The Sri Lankan cassava mosaic virus (SLCMV) was identified in Sri Lanka and linked to CMD in India [9].

The whitefly species predominantly spread CMD, particularly *B. tabaci* (Gennadius) (Hemiptera: Aleyrodidae). The *B. tabaci* whitefly is known to vector cassava mosaic begomoviruses and cassava brown streak ipomovirus, the causative agents of CMD, the cassava brown streak disease, and the SLCMV [7,9,11]. *B. tabaci,* the only vector of cassava geminiviruses, is spread predominantly by virus-infected cuttings. In Africa, these vectors have been shown to reduce cassava yields by significantly 35–60%. While nine CMV species have been reported in Africa and the Indian Ocean islands, only two are found in Asia: Indian cassava mosaic virus and the SLCMV, with the latter reported exclusively in Southeast Asia. The SLCMV has caused devastating yield losses in Thailand, with reductions of up to 80% [12,13,14].

Whiteflies, specifically *B. tabaci*, are tiny pests commonly found in tropical and subtropical areas [15]. These sap-sucking insects act as vectors for viruses, significantly harming various host plants. They significantly threaten cassava production in tropical regions [16]. Besides transmitting viruses, whiteflies directly harm cassava plants by feeding on the phloem of their leaves, resulting in leaf discoloration, shedding, and a potential yield reduction of up to 50% in vulnerable varieties [17]. Furthermore, the sugary substance (honeydew) excreted by whiteflies promotes the growth of a sooty mold, further impeding the photosynthetic capability of cassava plants.

CMD can be managed through the multiplication and distribution of disease-free stem cuttings. The main management for the SLCMV is host plant resistance against viruses through CMD used throughout and introduced to Thailand. However, producing cassava resistant to the SLCMV via conventional breeding methods is challenging due to the high heterozygosity and inbreeding depression in elite cultivars [18]. Additionally, the excessive use of chemical pesticides for whitefly control harms the ecosystem and raises production costs, proving uneconomical for small-scale farmers [19]. Consequently, developing host plant resistance to insect vectors emerges as one of the most effective strategies for controlling vector-borne viral diseases [20]. The successful transmission of plant viruses by insect vectors hinges on their behavior and dispersal capabilities. Whiteflies are particularly pivotal in virus transmission, making their host plant selection process critically important. Insect herbivores rely on sensory mechanisms, such as olfactory receptors for plant volatiles and gustatory and mechanoreceptive sensillae for feeding and oviposition, to identify suitable host plants [21,22,23]. Plant defenses, including trichome density and leaf surface characteristics, play essential roles in repelling pests [24,25,26]. As hemipteran vectors of plant viruses, whiteflies adopt steps to locate and identify suitable host plants for settling and feeding, engaging in sustained phloem-sap ingestion once they deem the host plant appropriate [27].

Whitefly-stylet activities—particularly penetration into phloem sieve elements—are closely associated with transmitting phloem-restricted persistent viruses. Most hemipterans prefer settling on the abaxial side of the leaf upon landing [28]. For instance, *Myzus persicae* aphids tend to settle on the abaxial leaf surface, and so do *B. tabaci* crawlers when given a choice [29]. After touching the plant surface, *B. tabaci* evaluates host plant quality through labial dabbing and probing with its piercing mouthparts. The feeding behavior of piercing–sucking insects such as whiteflies can be closely monitored using the electrical penetration graph (EPG) technique. This method, pioneered by McLean et al., [30] and refined by Tjallingii [31,32], has proven instrumental in understanding the feeding behavior of whiteflies such as *Trialeurodes vaporariorum* and *B. tabaci* [33,34,35,36,37,38,39,40]. The waveform patterns of *B. tabaci* were categorized by amplitude, relative voltage level, R/emf origin, frequency, and the context of the waveform as non-probing (Np), stylet pathway (C), phloem salivation (E1), phloem ingestion (E2), intracellular puncture—potential drop (Pd), and xylem feeding (G). The EPG technique involves creating an electric circuit that includes the insect and the plant, where voltage fluctuations are associated with specific stylet activities. This technique helps pinpoint the tissues in which plant resistance factors operate [41].

Understanding whitefly preferences for feeding on different cassava cultivars is crucial for developing resistant varieties and mitigating the impact of viral diseases on cassava production. This study examines the feeding behavior and settling preferences of *B. tabaci* across diverse cassava cultivars and evaluates whitefly spread among these cultivars. The insights gained will contribute to developing whitefly-resistant cassava varieties, ultimately bolstering cassava production and mitigating the impact of viral diseases on agricultural livelihoods.

## 2. Results

### 2.1. EPG Waveform Characteristics and Feeding Behavior of Bemisia Tabaci Whiteflies on Different Cassava Cultivars

#### 2.1.1. EPG Waveform Characteristics

The six probing waveforms identified in this study were Np, C, E1, E2, Pd, and G, which provide a comprehensive overview of whitefly feeding behavior. The waveform patterns, types, and characteristics are depicted in Figure 1. These waveform characteristics align with the findings of previous studies and reaffirm the complexity of whitefly interactions with host plants [36,37,38,40,42,43,44,45].

#### 2.1.2. Feeding Behavior of *Bemisia tabaci* Whiteflies on Different Cassava Cultivars

The feeding behavior of *B. tabaci* on six cassava cultivars was analyzed using the EPG technique. The frequency and duration of the six probing waveforms—Np, C, E1, E2, Pd, and G—were compared among the cultivars.

The TPD and TWD of adult whiteflies were calculated and compared between the six cassava cultivars (Figure 2). The data indicated that the adult whiteflies spent most of their time in the Np phase on the cassava leaf tissue, followed by the E1 waveform. Notably, the percentage of time spent in the Np phase was significantly longer on Huaybong 80, Kasetsart 50, Rayong 9, and Rayong 72 cultivars, averaging around 70%, compared to Rayong 5 and CMR 89 cultivars. The TPD for adult whiteflies revealed that the whiteflies spent more than 50% of the recording time on CMR 89 cultivars in probing. Additionally, CMR 89 showed higher total probe durations in the E1 and E2 waveforms (Figure 2).

#### 2.1.3. Number of Waveform Events per Insect (NWEI)

The NWEI for the Pd waveform did not show significant differences among the six cassava cultivars. However, the number of Np (F5,182 = 1.504, *p* = 0.023), C (F5,276 = 1.458, *p* = 0.024), E1 (F5,287 = 1.530, *p* = 0.022), E2 (F5,195 = 1.640, *p* = 0.057), and G (F5,105 = 1.788, *p* = 0.018) waveform events per insect revealed significant differences among the six cassava cultivars. Adult *B. tabaci* whiteflies spent more time on the E1 (phloem salivation) waveform in Huaybong 80, Rayong 9, and CMR-89 cultivars. They also spent more time on the E2 (phloem ingestion) waveform in Rayong 5, Huaybong 80, CMR-89, and Rayong 9 cultivars, similar to the G waveform (Figure 3).

#### 2.1.4. Waveform Duration per Event per Insect (WDEI)

The WDEI for the C, Pd, and G waveforms did not show significant differences among the six cassava cultivars. However, the duration of the Np, E1, and E2 waveforms differed significantly (Figure 4). Adult *B. tabaci* whiteflies spent a longer duration per event on the Np waveform when fed on Huaybong 80 (3981.04 ± 148.53 s) and Kasetsart 50 (3818.58 ± 149.98 s). Additionally, the WDEI for E1 and E2 waveforms was longer in CMR-89 (625.61 ± 109.68 s for E1; 302.04 ± 12.50 s for E2), Rayong 72 (554.73 ± 144.1 s for E1; 160.87 ± 10.09 s for E2), and Rayong 5 (480.96 ± 104.73 s for E1; 133.54 ± 73.76 s for E2) than in Rayong 9, Kasetsart 50, and Huaybong 80 (Figure 4).

#### 2.1.5. Waveform Duration per Insect (WDI)

The WDI for the Pd and G waveforms did not show significant differences among the six cassava cultivars. However, the WDI for the Np, C, E1, and E2 waveforms were significantly different (F5,182 = 2.841, *p* = 0.037 for waveform Np; F5,276 = 2.026, *p* = 0.011 for waveform C; F5,287 = 3.972, *p* = 0.009 for waveform E1; and F5,195 = 1.981, *p* = 0.045 for waveform E2). The Np waveform in adult *B. tabaci* whiteflies had a shorter duration in CMR-89 (4671.56 ± 357.59 s) than in other cassava cultivars. However, the E1 and E2 waveforms had longer durations in CMR-89 (4752.75 ± 199.02 s for E1; 1068.16 ± 139.95 s for E2) than in other cassava cultivars (Figure 5).

### 2.2. Adult Bemisia tabaci Whitefly Settling Under Free Choice

An analysis of variance was conducted to assess the settling preferences of the *B. tabaci* whiteflies on different cassava cultivars under field conditions for nine weeks. The results indicated significant differences in the settling behavior of the whiteflies among the cultivars (*p* ≤ 0.05) during the first week. Specifically, Rayong 9 and Huaybong 80 had the minimum number of adult whiteflies, while CMR-89 and Rayong 5 had the maximum (F5,30 = 16.140, *p* < 0.001). In the second week, the minimum number of adult whiteflies varied significantly among cultivars, particularly on Kasetsart 50 and Huaybong 80 (F5,30 = 1.0533, *p* = 0.043). From weeks 3 to 9, the minimum number of adult whiteflies remained significantly lower on Kasetsart 50 and Huaybong 80 compared to other cultivars (Figure 6). Conversely, CMR-89 and Rayong 5 consistently had the maximum number of adult whiteflies during weeks 3, 4, 5, 7, and 8. Specifically, significant differences were observed in week 3 (F5,30 = 3.933, *p* = 0.024), week 4 (F5,30 = 12.582, *p* < 0.001), week 5 (F5,30 = 6.060, *p* = 0.005), week 7 (F5,30 = 13.147, *p* < 0.001), and week 8 (F5,30 = 7.793, *p* = 0.002) (Figure 6). After four months, the CMR-89 cultivar showed moderate to severe SLCMV disease symptoms, which were severe in 30% of total cassava recorded. This suggests a correlation between the high whitefly settling preference on CMR-89 and the increased severity of the SLCMV in this cultivar.

### 2.3. Trichome Density and Size

The morphology of leaves of different cassava cultivars was observed by an SEM, which revealed that each cultivar’s characteristic was non-glandular trichomes (Figure 7). The trichome numbers per microscopic length scale of 50 and 100 µm^2^ of the SEM for the six cassava cultivars (Figure 7, Table 1) differed significantly (F5,59 = 56.89; *p* = < 0.01). The CMR 89 cultivar had the highest number of trichomes (256 ± 12.08 per 100 µm^2^), while the Huaybong 80 had the lowest (128.50 ± 14.19 per 100 µm^2^). For 50 µm of the SEM, the six cassava cultivars differed significantly (F5,59 = 34.669; *p* ≤ 0.01). The CMR 89 and Rayong 5 cultivars had the highest number of trichome density at 75.00 ± 5.29 and 50.83 ± 3.46 per 50 µm^2^, respectively, while the Huaybong 80 had the lowest (33.25 ± 4.66 per 50 µm^2^). The average size of each cassava cultivar leaf’s trichomes was significantly different (*p* ≤ 0.01). The Kasetsart 50 cultivar had the largest trichomes (11.34 ± 0.29 µm^2^), followed by the Huaybong 80 and Rayong 9 cultivars, which were 10.39 ± 0.25, and 10.02 ± 0.20 per 50 µm^2^ (F5,59 = 26.236, *p* ≤ 0.01), respectively (Table 1). The study highlights the variation in trichome characteristics among different cassava cultivars, indicating potential differences in their physiological and ecological traits.

### 2.4. The Correlation Analysis Between the EPG Parameters, Settling Preference of Whitefly and Trichome Density and Size

The correlation analysis conducted in this study aimed to explore the relationship between various parameters of the EPG activity of an adult *B. tabaci* whitefly and the settling preference on cassava cultivars in conjunction with trichome density and size (Table 2).

Conversely, the duration of settling preference and trichome density were positively correlated. Specifically, weeks 1, 5, 8, and 9 exhibited strong positive correlations with trichome density per 100 µm (r = 0.90*, r = 0.88*, r = 0.90*, r = 0.83, respectively) and per 50 µm (r = 0.90*, r = 0.83*, r = 0.83*, r = 0.78, respectively). Additionally, positive correlations were observed between settling preference and certain EPG waveform parameters, such as E1 and E2 duration per event per insect and E2 and G duration per insect. Additionally, the EPG waveform positively correlated with the trichome density. These findings suggest that higher trichome density, particularly in the CMR 89 cultivar, may not deter the feeding and settlement of adult whiteflies.

## 3. Discussion

### 3.1. EPG Waveform Characteristics and Feeding Behavior of Bemisia tabaci Whiteflies on Different Cassava Cultivars

Trichome density, size, and EPG waveform characteristics significantly influence the interaction between cassava plants and whiteflies (*B. tabaci*). Through the analysis of EPG waveform characteristics, six distinct probing waveforms were identified, including Np, C, E1, E2, Pd, and G, with the findings of previous studies and reaffirmed the complexity of whitefly interactions with host plants [36,37,38,40,42,43,44,45]. The waveforms provide insights into the feeding behavior of whiteflies on different cassava cultivars. The study revealed that adult whiteflies predominantly spent their time in the Np phase, followed by the E1 waveform, indicating a pattern of probing and feeding behavior. Significant differences were observed in the TPD and TWD among the six cassava cultivars. Specifically, CMR 89 cultivars exhibited longer total probe durations in the E1 and E2 waveforms than other cultivars, suggesting variations in whitefly feeding behavior across cultivars. Moreover, the NWEI and the WDEI varied significantly among the cassava cultivars. While the number of Pd waveform events did not show significant differences, other waveforms such as Np, C, E1, E2, and G exhibited significant variations. These differences reflect the distinct responses of whiteflies to the cassava cultivars, potentially influenced by factors such as leaf morphology and chemical composition.

The analysis of feeding behavior revealed that adult whiteflies predominantly remained in the Np phase, especially on Huaybong 80, Kasetsart 50, Rayong 9, and Rayong 72 cultivars. This finding suggests that these cultivars may possess certain deterrent properties or structural defenses that inhibit whitefly probing and feeding. In contrast, the CMR 89 and Rayong 5 cultivars exhibited higher probing activities, indicating a lower resistance to whitefly feeding. Moreover, the significant differences in the NWEI (Np, C, E1, E2, G) among the six cassava cultivars underscore the variability in whitefly feeding behavior influenced by plant genotype. Notably, the higher frequency of E1 and E2 waveforms in certain cultivars, such as CMR 89, indicates a preference for these plants, possibly due to easier access to phloem sap. In CMR 89, the shorter duration of the Np waveform and longer durations for E1 and E2 waveforms reinforce the susceptibility of this cultivar to whitefly feeding. The significant differences in WDI for the Np, C, E1, and E2 waveforms among the cultivars suggest that certain cassava genotypes are more conducive to whitefly feeding, which has implications for virus transmission dynamics [46,47]. The EPG technique has been used to study insect-feeding behavior in relation to the transmission of pathogens [48,49]. The prolonged duration in the non-probing phase on specific cultivars indicates a potential resistance mechanism, making these cultivars less suitable for whitefly infestation. This finding is significant for cassava breeding programs aiming to enhance pest resistance.

### 3.2. Adult Bemisia tabaci Whitefly Settling Under Free Choice Tested

During the first week, Rayong 9 and Huaybong 80 exhibited the lowest number of adult whiteflies, indicating an initial deterrence effect. In contrast, CMR-89 and Rayong 5 attracted the highest number of whiteflies. These differences were statistically significant, highlighting that the cultivars’ inherent characteristics influence whiteflies’ initial selection of host plants. In the second week, Kasetsart 50 and Huaybong 80 continued to show a significantly lower number of whiteflies. From weeks 3 to 9, Kasetsart 50 and Huaybong 80 consistently had fewer whiteflies, indicating sustained deterrence. The persistent lower whitefly numbers on these cultivars suggest they have long-term resistance traits, making them less attractive or hospitable to whiteflies.

Conversely, CMR-89 and Rayong 5 consistently had the highest number of whiteflies during the same period, particularly in weeks 3, 4, 5, 7, and 8. The significant differences observed in these weeks indicate that these cultivars are more attractive to whiteflies, possibly due to higher phloem availability or fewer physical or chemical deterrents. The high whitefly settling preference on CMR-89 notably correlates with severe SLCMV disease expression in 30% of the plants after four months. This correlation highlights the significant role of whitefly preference in the epidemiology of SLCMV disease. However, whiteflies transmitted infections were less severe than those originating from infected cuttings, with many infected plants remaining asymptomatic. Certain genotypes, including CMR-89 and Ryong 11, were also susceptible to CMD [14]. The increased settling on CMR-89 likely facilitated more effective virus transmission, leading to higher disease severity. This relationship between whitefly behavior and SLCMV severity suggests that managing whitefly populations is crucial for controlling the disease. The findings from this study have significant implications for cassava breeding programs. Cultivars such as Kasetsart 50 and Huai Bong 80, which consistently showed lower whitefly numbers, should be prioritized for breeding and cultivation in areas prone to whitefly infestations and the SLCMV. The inherent resistance of these cultivars to whitefly settling could reduce the incidence of vector-borne diseases, thereby improving crop health and yield. Conversely, cultivars such as CMR-89 and Rayong 5, which attracted more whiteflies and exhibited higher SLCMV severity, may require additional management strategies or could be modified through breeding to incorporate resistance traits. Understanding the specific characteristics conferring resistance in Kasetsart 50 and Huai Bong 80 could guide the development of new, more resilient cassava varieties. The settling preference analysis further highlighted significant differences in whitefly settling behavior among the cassava cultivars. For instance, cultivars such as Rayong 9 and Huaybong 80 consistently exhibited fewer settling whiteflies than CMR 89 and Rayong 5. The observed variations in settling preference may be attributed to factors such as leaf trichome density and size. Indeed, the correlation analysis revealed intriguing relationships between whitefly settling preference, trichome density, and size. A negative correlation was observed between the number of settling whiteflies and trichome size, suggesting that larger trichomes may deter whitefly settling. Conversely, a positive correlation was found between settling preference duration and trichome density, particularly in cultivars such as CMR 89.

The SLCMV transmitted by the insect vector *B. tabaci* causes CMD in cassava plants [12]. Whiteflies select host plants upon landing, utilizing their stylets to create brief probes in the plant’s epidermis for feeding and oviposition, a process not guided by olfactory cues [50]. This study exposed six cassava plants to field conditions infested with whiteflies. Interestingly, the initial leaf position of all six cassava cultivars showed lower whitefly infestation than the other leaf positions did. Variations in leaf position were found to impact the content of cyanide, amino acids, and crude protein. Specifically, cyanide and amino acid levels were higher in the upper leaf positions compared to the lower ones, while crude protein content exhibited the opposite trend, being higher in the lower leaves [51]. Cyanides and amino acids possess direct toxicity towards insect herbivores, with amino acids enabling nitrogen storage in a form inaccessible to them [52,53]. This inherent toxicity contributes to the aversion of whiteflies towards infesting the first leaf position more than others.

### 3.3. Trichome Density and Size 

Trichomes affect insect behavior, influencing egg laying, shelter seeking, and feeding [54,55,56,57]. The results from the no-choice infestation test revealed differing whitefly infestation percentages among the six cassava cultivars. Notably, the Rayong 72 cultivar exhibited the lowest whitefly infestation compared to the Huaybong 80 and CMR-89 cultivars. The SEM of trichomes across all cassava cultivars revealed distinct differences. Rayong 72 displayed a smaller trichome size but higher trichome density. Trichome density has contributed to whitefly resistance in various crops, such as cotton, eggplant, tomato, chilies, soybean, and cucumber [58,59,60,61,62]. The non-glandular trichomes on the abaxial surface of all six cassava varieties serve as a defense mechanism against insect attacks, effectively restricting insect movement and as a barrier protecting the plant’s epidermal layer from damage [57,63]. Consequently, the preference of whiteflies for egg laying, shelter seeking, or feeding on these plants is reduced.

The sparse distribution of trichomes or the thinness of the epidermis may have facilitated whitefly feeding, a crucial factor influencing the host plant preference of insect herbivores [64]. Whitefly feeding behavior on the host plant involves intracellular penetration by the insect’s stylet, which is highly efficient for transmitting viruses to the host plant [65]. The SLCMV poses a significant threat to crop yield, leading to substantial economic losses and food insecurity [18]. This study investigated whitefly feeding behavior using EPG. Initially, the whiteflies traversed the abaxial surface of cassava leaves, generating an Np. Subsequently, the whitefly’s stylet penetrated the epidermis and parenchyma tissue (waveform C). Upon reaching the phloem sieve tube, the insect released saliva (E1) and ingested phloem sap (E2). Across all cassava cultivars, the whitefly’s stylet penetration of intracellular vascular tissue (Pd: potential drop) lasted only briefly. The Rayong 72 cultivar exhibited a short duration of phloem ingestion and a low number of ingestion events, potentially contributing to the limited spread of the SLCMV. Conversely, the Huaybong 80 cultivar showed an extended total duration of phloem salivation, increasing the likelihood of SLCMV transmission into the phloem sieve.

### 3.4. The Correlation Analysis 

According to the findings in Table 2, a negative correlation was observed between the number of adult whiteflies settling on cassava cultivar leaves and the size of the trichomes. These findings are consistent with those of [66,67,68], who observed a negative correlation between trichome density on eggplant leaves and resistance to *B. tabaci*. Similar results were also reported by [69] in cucumbers, [70] in tomatoes, [71] in soybeans, and [72] in cotton plants. Furthermore, negative correlations were also identified between the settling preference of adult whiteflies and the number of penetration events, Np waveform, the waveform C duration per event per insect WDEI, and the Np waveform duration per insect. These results suggest that larger trichomes may deter adult whiteflies from settling, while certain EPG parameters indicative of feeding behavior were also negatively correlated with settling preference.

These findings underscore the complex interplay between trichome characteristics, EPG waveform parameters, and whitefly settling preference on cassava cultivars. Understanding these relationships is crucial for devising effective pest management strategies and breeding programs to enhance cassava resistance to whiteflies. Further research is warranted to elucidate the underlying mechanisms driving these correlations and their implications for cassava production and crop protection.

## 4. Materials and Methods

The experiment was conducted inside a laboratory and field at the Suranaree University of Technology (14°58′14.38″ N 102°06′7.06″ E), Nakorn Ratchasima, Thailand.

### 4.1. Source of Planting Materials

Six cassava cultivars (*Manihot esculenta* Crantz)—namely, Rayong 5, Rayong 9, Rayong 72, Kasetsart 50, Huaybong 80, and CMR-89—were sourced from Nakhon Ratchasima, Thailand, for the screening of antixenosis resistance to the *B. tabaci* whitefly. The DNA of each cassava plant was extracted from the auxiliary buds of both ends of the plants to confirm disease and non-disease status, using polymerase chain reaction (PCR) with specific SLCMV primers [14]. The DNA extraction followed the protocols of [73,74], and [75], with quality and quantity assessed via spectrophotometry (Thermo Scientific™ NanoDrop™ 2000, Thermo Fisher Scientific, Waltham, MA, USA). The presence of the SLCMV was monitored by PCR, using AV1 gene-specific primers: AV1 forward (5′-GTT GAA GGT ACT TAT TCC C-3′) and AV1 reverse (5′-TAT TAA TAC GGT TGT AAA CGC-3′). PCR amplification involved a 25 μL reaction volume with 1XPCR buffer (PCR Biosystems, London, UK), 0.2 μM of each primer, and approximately 50 ng of the DNA template. The thermal cycling conditions were as follows: initial denaturation at 94 °C for 5 min, followed by 35 cycles of denaturation at 94 °C for 40 s, annealing at 55 °C for 40 s, extension at 72 °C for 40 s, and final elongation at 72 °C for 5 min. The PCR was performed using a Bio-Rad iCycler (Bio-Rad Laboratories, Hercules, CA, USA). DNA gel electrophoresis of PCR amplification from cassava samples (Appendix A). The non-disease cassavas were used for propagation and examined electropenetrography recording and whitefly *B. tabaci* settling experiment. The cassava stems were cut to 15–18 cm lengths, with 20 plants gathered in each cultivar. Six cassava plants were planted in 11 × 15 cm^2^ pots containing a mix of sandy loam soil and manure (1:1 ratio). The plants were watered daily, and no insect- or disease-control chemicals were applied. When the plants reached six weeks of age, they were used for the settling experiments.

### 4.2. Whitefly Materials

*B. tabaci* colonies were collected from Nakhon Ratchasima Province, Thailand, cassava fields. These were maintained under laboratory conditions at 28 ± 2 °C, 60–65% RH, with a 12:12 (L/D) photoperiod. Susceptible CMR-89 cassava cultivars were used for mass rearing. Twenty cassava plants were potted in 5-inch diameter plastic pots and placed in BugDorm-6E620 Insect-Rearing Cages (W60 × D60 × H120 cm^3^, MegaView Science Co., Ltd., Taichung, Taiwan). Two cages, each containing ten pots, were used, and plants were replaced every two weeks. Adult whiteflies within 24 to 48 h after emerging were used for experiments.

### 4.3. Electropenetrography Recording

Feeding-behavior experiments were conducted using a Giga-8 DC electropenetrography (EPG) system (EPG Systems, Wageningen, The Netherlands) with an input resistance of 10^9^ Ω (1 GΩ) and adjustable plant voltage. The cassava plant and the insect with the EPG probe were placed within a Faraday cage (1.5 × 2 × 1.5 m^3^) to block electrical noise. The system was installed in a temperature-controlled room at 26 ± 2 °C and 60 ± 5% RH. EPG signals were recorded by Stylet+ software (v01.34), adjusting the signal range from −5 to +5 V, and displayed on a computer [40]. Adult whiteflies were placed in a glass vial and immobilized on ice for 5–10 s before being connected to a gold wire electrode (2.5 cm long, 12 μm diameter) (Appendix A) (EPG Systems, Wageningen, The Netherlands), with water-based silver glue (Wageningen University) on the insect’s pronotum. The other end of the wire was attached to a copper electrode (1 mm diameter, 2.5 cm long) (Sigmund Cohn Corp, Mt Vernon, NY, USA), which was connected to the EPG probe. The whitefly was placed on the abaxial leaf surface fixed with parafilm. Signal adjustments were made if needed, and data were recorded using Stylet+d software (v01.34). Each whitefly was observed for 3 h daily, with 30 whiteflies tested per cultivar in a completely randomized design. Annotated waveforms (non-probing, stylet pathway, phloem salivation, phloem ingestion, intracellular puncture, and xylem feeding) were analyzed using EPG Systems software and a modified Ebert 3.0 program in SAS Enterprise Guide 7.1, SAS 9.4 statistical software.

EPG data recorded by EPG Systems Stylet+d were manually annotated using EPG Systems Stylet+a software (v01.34). The annotated waveforms were non-probing (Np), stylet pathway (C), phloem salivation (E1), phloem ingestion (E2), intracellular puncture—potential drop (Pd), and xylem feeding (G). The waveform patterns were categorized by amplitude, relative voltage level, R/emf origin, frequency, and the waveform context as described in the previous EPG studies of *B. tabaci* [36,37,38,40,42,43,44,45]. Annotation files were then directly passed to a modified version of the Ebert 3.0 program in SAS Enterprise Guide 7.1, SAS 9.4 statistical software (SAS Institute, Cary, NC, USA) for further analysis, which produces the same parameters as the popular Sarria Excel workbook [76,77].

### 4.4. Whitefly Bemisia tabaci Settling Under Free Choice in Field Condition

The free choice test used six cassava plants (Rayong 5, Rayong 9, Rayong 72, Kasetsart 50, Huaybong 80, and CMR-89 cultivars). The experiment was conducted in a randomized complete block design with 40 × 35 m^2^ pot dimensions, employing five replications per cultivar. Each treatment consisted of 30 cassava plants. Cassava stems were planted, measuring 15–18 cm in length with approximately four nodes each. The planting was conducted in mini-field conditions at the Suranaree Farm within the Suranaree University of Technology. A 1 × 0.8 m^2^ planting distance was maintained between the cassava plants. The white net greenhouse covered the field (40 meshes, 1.3 mm/0.05′, aperture). To mitigate the impact of weeds and potential hosts of pests, manual weeding using hand hoes was performed. No plant protection measures (such as pesticide applications) were implemented throughout the trial. When the plants were eight weeks old, 250 adults of the whitefly species *B. tabaci* were introduced into the center of the white net greenhouse where the test plants were situated. Adult whitefly infestation levels were assessed by counting and recording the number of *B. tabaci* adults settling on each cassava plant.

For accurate findings, whitefly (*B. tabaci*) infestation was meticulously assessed. The number of whiteflies settled per treatment was divided by the total number of whiteflies settled per cassava plant within each replication (20 cassava plants). This calculation determined the proportion of whiteflies settled on each treatment (three times). Whitefly infestation on cassava plants was monitored weekly under field conditions for a period ranging from 1 to 9 weeks. Upon completion of the experimental period, cassava plants were maintained for up to 4 months. During this period, the plants were observed for symptoms indicative of SLCMV infection. The cassava plants that showed SLCMV symptoms were DNA extracted, and their disease status was determined using the PCR technique with specific SLCMV primers.

### 4.5. Trichome Size and Density

The morphology of the abaxial surface of the six cassava leaves (Rayong5, Rayong9, Rayong72, Kasetsart50, Huaybong80, and CMR-89) was analyzed for trichome density and size. Samples from the fourth leaf of 8-week-old plants were fixed in 37% formaldehyde overnight, washed in phosphate-buffered saline, and dehydrated in an ethanol series (70%, 80%, 95%, and 100%) [78,79]. Samples were critical-point dried and coated with gold-palladium [57,79]. The fixed surfaces of the leaves were then observed under a cutting-edge scanning electron microscope (SEM) (SEM™, FEI, Quanta450, Eindhoven, The Netherlands). Trichome density was counted manually per 100 and 50 µm^2^, with ten biological replications and five images per leaf. Trichome sizes were measured for each treatment with 15 replications per cultivar.

### 4.6. Statistical Analysis

EPG waveforms related to feeding-behavior events were characterized: Np, C, E1, E2, Pd, and G; the variables total probing duration (TPD), total waveform duration (TWD), number of waveform events per insect (NWEI), waveform duration per event per insect (WDEI), and waveform duration per insect WDI) were also calculated (mean ± standard error), as described by [80]. The variables were compared using the Tukey–Kramer test at *p* < 0.05. Free choice test data were analyzed using the Mann–Whitney U-test at *p* < 0.05. All analyses were performed with SAS V9.4 software. The Tukey–Kramer test conducted mean comparisons for trichome density and size. Pearson correlation analysis was conducted on mean values of whitefly feeding behavior, insect settling, and trichome density and size using Origin Pro 2024 software, with significance set at *p* < 0.05 [81].

Data from the free choice in field condition, calculated as the mean member of whiteflies counted in the antixenosis resistance screening of six cassava cultivars at different time intervals (subjected to the Mann–Whitney U-test at *p* < 0.05 significance level), were computed using the SAS V9.4 software. All statistical analyses were calculated using Statistic Analysis System V9.4 software (SAS Institute, Inc., Cary, NC, USA). The mean was compared using the Tukey–Kramer test for trichome density and size.

For correlation analysis (Pearson correlation), conducted by [57,82,83] on the mean values of whitefly feeding behavior, number of settled insects, and trichome density and size to identify the relationships between parameters in this study. The differences were considered significant at a probability level of 5%, using Origin Pro 2024 software (OriginLab Corporation Northampton, MA, USA) [81].

## 5. Conclusions

In summary, the results indicate that trichome characteristics play a significant role in whitefly settling preference on cassava cultivars, with larger trichomes and higher trichome density correlated to reduced settling behavior. These findings have implications for pest management strategies, highlighting the potential of trichome-based resistance mechanisms in cassava breeding programs.

## Figures and Tables

**Figure 1 plants-13-03218-f001:**
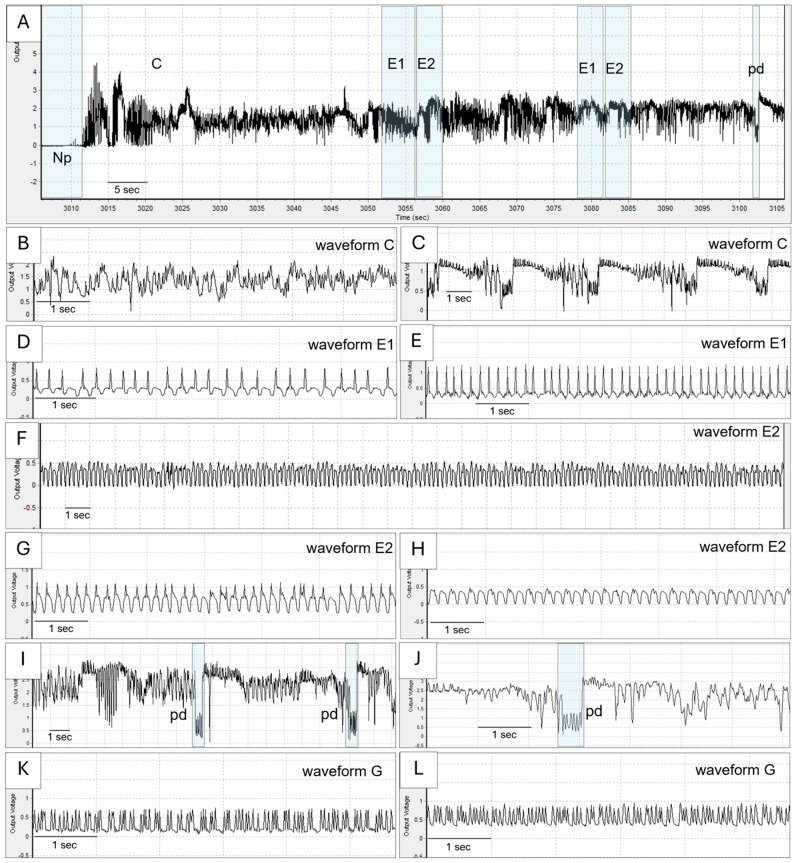
EPG waveforms of *Bemisia tabaci* whitefly were observed in the study. (**A**) The overview of EPG waveforms and details for 100 s with non-probing (Np), stylet pathway (**C**), phloem salivation (E1), phloem ingestion (E2), and potential drop (Pd) waveforms. Waveform C, (**B**,**C**). Waveform E1 (**D**,**E**). Waveform E2 (**F**–**H**). Waveform pd (**I**,**J**). Waveform G (**K**,**L**).

**Figure 2 plants-13-03218-f002:**
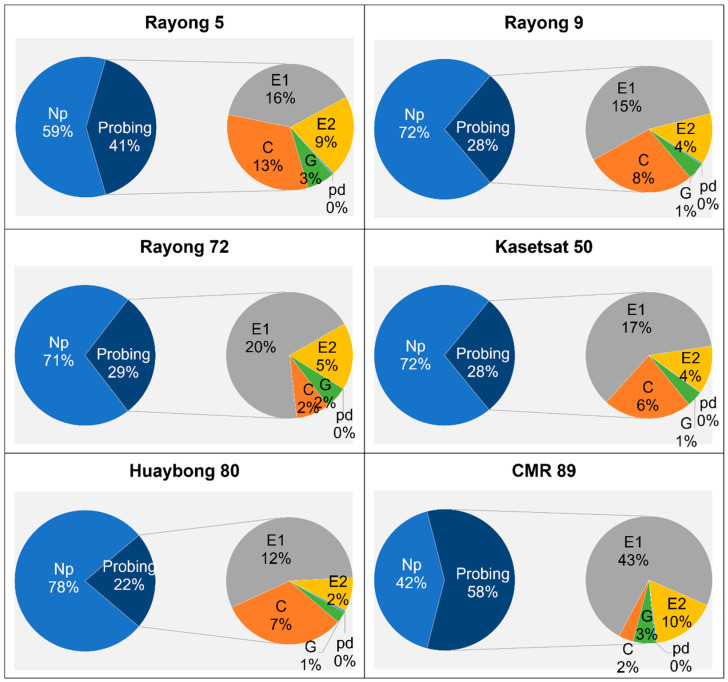
The total waveform duration (TWD) and the total probe duration (TPD) of adult *B. tabaci* whiteflies on different cassava cultivars during 3 h recording. Np, C, E1, E2, Pd waveform, and G waveform. Np: non-probing, waveform C: stylet pathway, waveform E1: phloem salivation, waveform E2: phloem ingestion, waveform Pd: intracellular puncture—potential drop, and waveform G: xylem feeding.

**Figure 3 plants-13-03218-f003:**
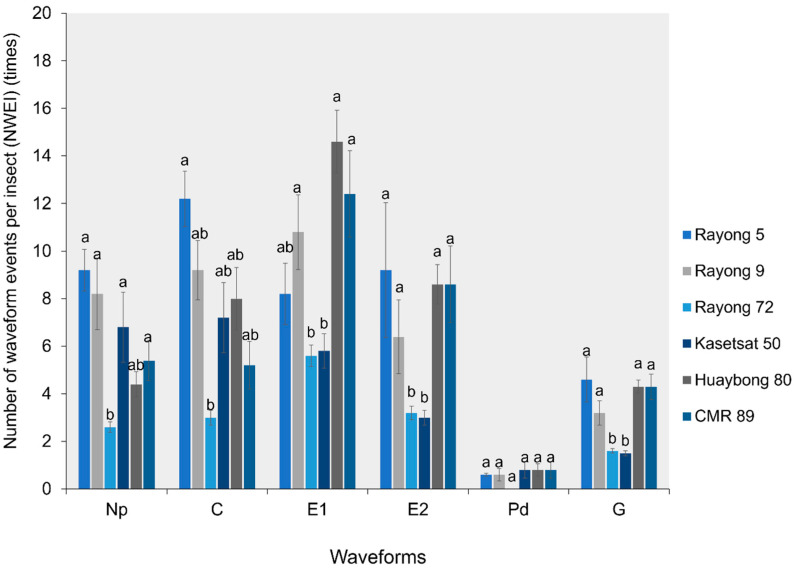
The number of waveform events per insect (NWEI) for all waveforms on six cassava cultivars. Bars (mean ± SE) with the same letter at the top within a waveform category are not significantly different at *p* = 0.05 (Tukey’s test): Np = non-probe; C = pathway phase; E1 = salivation phase; E2 = ingestion phase; Pd = potential drops; and G = xylem feeding.

**Figure 4 plants-13-03218-f004:**
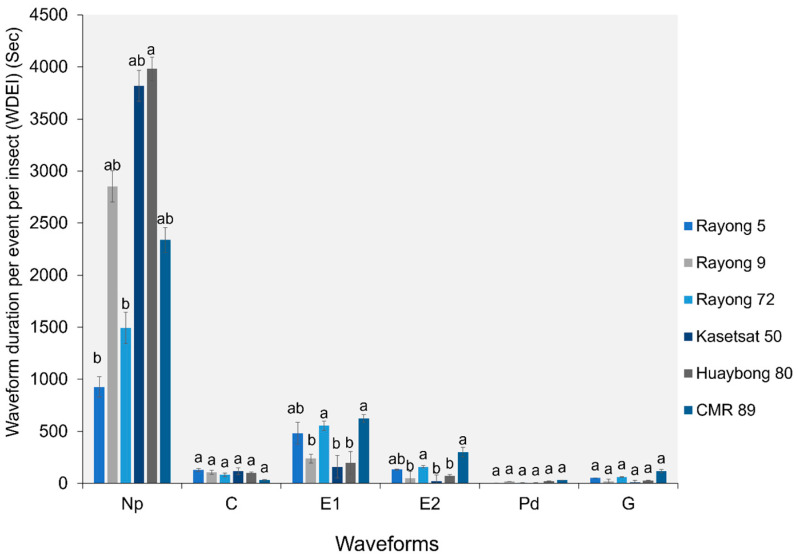
The waveform duration per event per insect (WDEI) for all waveforms on six cassava cultivars. Bars (mean ± SEM) with the same letter at the top within a waveform category are not significantly different at *p* = 0.05 (Tukey’s test): Np = non-probe; C = pathway phase; E1 = salivation phase; E2 = ingestion phase; Pd = potential drops; and G = xylem feeding.

**Figure 5 plants-13-03218-f005:**
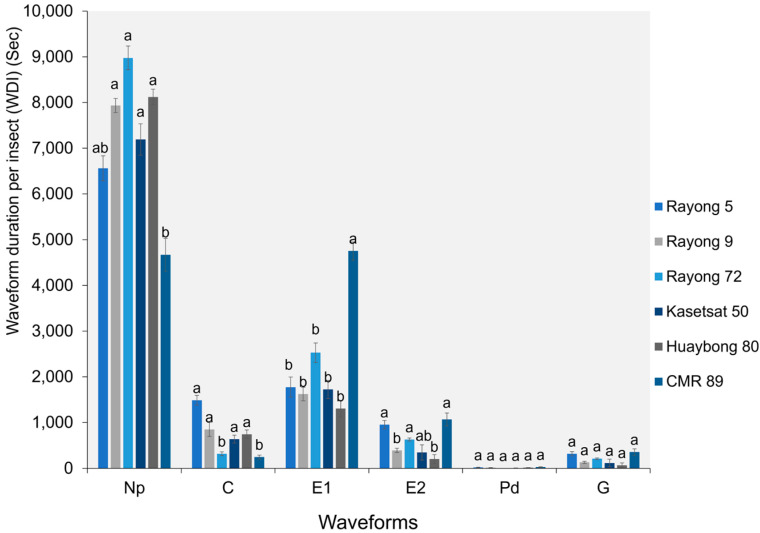
The WDI for all waveforms on six cassava cultivars. Bars (mean ± SEM) with the same letter at the top within a waveform category are not significantly different at *p* = 0.05 (Tukey’s): Np = non-probe; C = pathway phase; E1 = salivation phase; E2 = ingestion phase; Pd = potential drops; and G = xylem feeding.

**Figure 6 plants-13-03218-f006:**
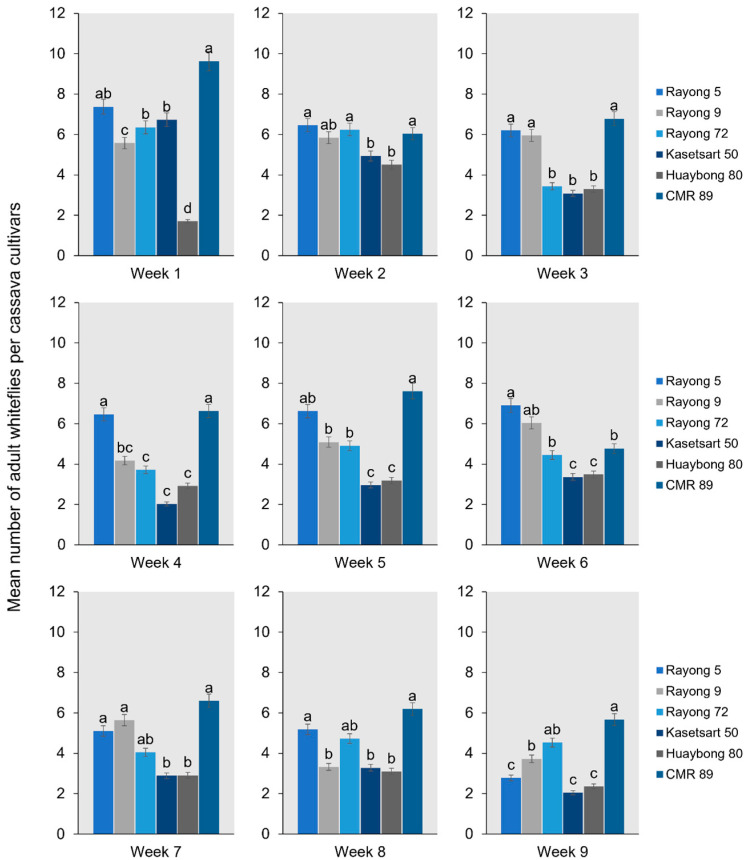
The mean number of adult whiteflies settled on the cassava cultivar was 1 to 9 weeks after release. Bars represent the mean percentage of whiteflies settled (mean ± SEM). Different letters above the bars indicate significant differences.

**Figure 7 plants-13-03218-f007:**
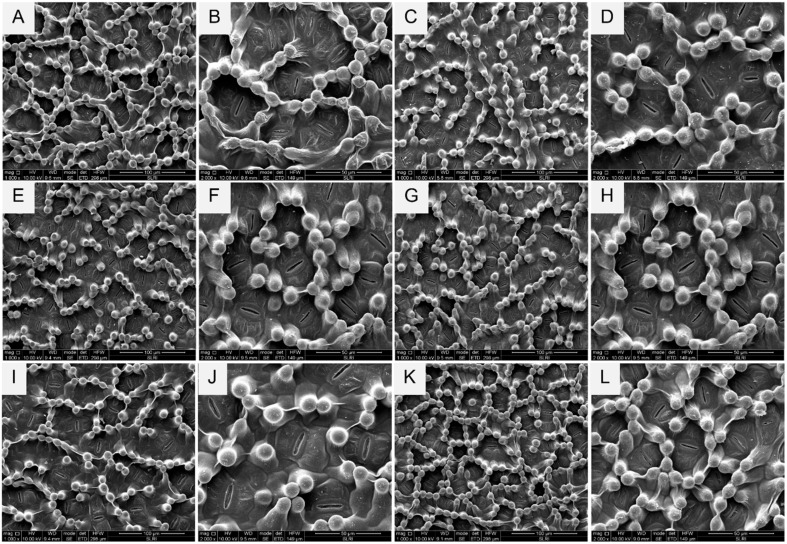
Trichomes of six cassava cultivar leaves with scanning electron micrograph of cassava leaf (**A**,**B**) Rayong 5, (**C**,**D**) Rayong 9, (**E**,**F**) Rayong 72, (**G**,**H**) Kasetsart 50, (**I**,**J**) Huaybong 80, and (**K**,**L**) CMR-89. (**A**,**C**,**E**,**G**,**I**,**K**): SEM of cassava leaves with 100 µm^2^, (**B**,**D**,**F**,**H**,**J**,**L**): SEM of cassava left with 50 µm^2^.

**Table 1 plants-13-03218-t001:** Mean ± SEM trichome density and trichome size of six cassava cultivars.

Cassava Cultivars	Trichome Density per 100 µm^2^				Trichome Density per 50 µm^2^				Size of Trichome (µm)			
Rayong 5	187.33	±	18.16	b ^1/^	50.83	±	3.46	b	9.76	±	0.27	c
Rayong 9	170.33	±	12.19	b	48.67	±	0.88	b	10.02	±	0.20	b
Rayong 72	180.67	±	15.61	b	47.00	±	3.06	b	9.80	±	0.26	c
Kasetsat 50	161.00	±	25.01	b	46.50	±	3.50	b	11.34	±	0.29	a
Huaybong 80	128.50	±	14.19	c	33.25	±	4.66	c	10.39	±	0.25	b
CMR 89	256.00	±	12.08	a	75.00	±	5.29	a	9.01	±	0.17	c

^1/^ Different letters indicate significant differences.

**Table 2 plants-13-03218-t002:** Correlation coefficients (Pearson correlation) and significance levels of the EPG parameters, the settling preference of whiteflies, and trichome density and size among six cassava varieties.

Parameters	Trichome per 100 µm	Trichome per 50 µm	Size of Trichome	Settling Preference of Adult Whiteflies on Cassava Cultivars After Different Infection Times								
				Week1	Week2	Week3	Week4	Week5	Week6	Week7	Week8	Week9
Number of Np waveform events per insect	0.04	0.11	0.10	0.23	0.21	0.55	0.33	0.24	0.67	0.32	−0.05	−0.35
Number of C waveform events per insect	−0.27	−0.23	0.15	−0.16	0.02	0.37	0.24	0.06	0.60	0.07	−0.20	−0.57
Number of E1 waveform events per insect	0.00	0.04	−0.33	−0.38	−0.37	0.31	0.22	0.13	−0.05	0.23	−0.04	0.12
Number of E2 waveform events per insect	0.23	0.22	−0.55	−0.04	0.10	0.64	0.68	0.52	0.45	0.46	0.35	0.10
Number of Pd waveform events per insect	0.01	0.13	0.22	−0.05	−0.54	0.22	0.03	−0.07	−0.16	0.02	−0.16	−0.33
Number of G waveform events per insect	0.23	0.22	−0.55	−0.04	0.10	0.64	0.68	0.52	0.45	0.46	0.35	0.10
Waveform Np duration per event per insect	−0.50	−0.39	0.64	−0.55	−0.95 **	−0.50	−0.72	−0.73	−0.74	−0.53	−0.72	−0.44
Waveform C duration per event per insect	−0.73	−0.72	0.65	−0.44	−0.17	−0.31	−0.37	−0.50	0.23	−0.52	−0.61	−0.87 *
Waveform E1 duration per event per insect	0.81 *	0.71	−0.86 *	0.68	0.80	0.52	0.78	0.84 *	0.37	0.65	0.95 **	0.81
Waveform E2 duration per event per insect	0.88 *	0.81 *	−0.87 *	0.64	0.55	0.56	0.76	0.83 *	0.17	0.68	0.93 **	0.86 *
Waveform Pd duration per event per insect	0.42	0.47	−0.49	0.05	−0.19	0.41	0.29	0.32	−0.16	0.48	0.22	0.55
Waveform G duration per event per insect	0.88 *	0.81 *	−0.87 *	0.64	0.55	0.56	0.76	0.83	0.17	0.68	0.93	0.86
Waveform Np duration per insect	−0.79	−0.83	0.45	−0.70	−0.23	−0.70	−0.68	−0.68	−0.21	−0.63	−0.67	−0.39
Waveform C duration per insect	−0.34	−0.34	0.16	−0.18	0.14	0.25	0.22	0.04	0.64	−0.03	−0.16	−0.60
Waveform E1 duration per insect	0.93 **	0.91 **	−0.71	0.75	0.42	0.51	0.61	0.72	0.01	0.67	0.83	0.87
Waveform E2 duration per insect	0.88 *	0.82 *	−0.78	0.84	0.80	0.74	0.91 **	0.94 **	0.55	0.75	0.97 **	0.64
Waveform Pd duration per insect	0.65	0.67	−0.64	0.43	0.25	0.83	0.81	0.74	0.41	0.72	0.61	0.38
Waveform G duration per insect	0.88 *	0.82 *	−0.78	0.84 *	0.80	0.74	0.91 **	0.94 **	0.55	0.75	0.97 **	0.64
Trichome density per 100 µm		0.98 **	−0.75	0.90 *	0.65	0.73	0.78	0.88 *	0.32	0.83	0.90 *	0.83
Trichome density per 50 µm			−0.67	0.90 *	0.57	0.73	0.73	0.83	0.29	0.83	0.83	0.78
Size of trichome				−0.47	−0.70	−0.75	−0.87 *	−0.90 *	−0.52	−0.84	−0.82	−0.85 *

Asterisks indicate significant difference (* *p* < 0.05, ** *p* < 0.01).

## Data Availability

Data can be provided upon request from the lead author.

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
