# Peer review of "Impact of Cassava Cultivars on Stylet Penetration Behavior and Settling of Bemisia tabaci Gennadius (Hemiptera: Aleyrodidae)"

_plants, 2024, doi:10.3390/plants13223218_

Round 1
Reviewer 1 Report
Comments and Suggestions for Authors
The manuscript by Pimkornburee et al. reports studies on feeding behavior of B. tabaci on cassava using electrical penetration graphs and by measuring density and size of abaxial leaf trichomes using SEM. The findings of this study, that trichome size and density correlates with whitefly settling behavior can guide breeding strategies for whitefly resistance in cassava. A reduced number of whiteflies settling on cassava then has implications for transmission of viruses causing cassava mosaic disease, like SLCMV. All current Thai varieties are susceptible to SLCMV and deterring whiteflies from settling on cassava by breeding for a particular trichome size and density is very valid. Thus, the report is on an interesting subject, the manuscript is well written and the research is well described.
However, there are several shortfalls the authors need to address before this article can be recommended for publication. They mostly concern the virus aspects of the work which need to be thoroughly revised and speculative statements removed, particularly when nothing is known about the experimental plants' susceptibility, virus titer etc (e.g. line 364 ….potentially contributing to the limited spread of SLCMV).
Major
1. Virus detection by PCR is described in M&M but there are no associated results/ discussion.
2. In 2.2. there is mention of severe virus infection in 30% in CMR-89 that is 6 plants of 20?
a. Other varieties are also susceptible to SLCMV and but no infections in other varieties were reported?
b. How can it be excluded that the CMR-89 plants were not infected from cuttings? Where did the virus come from? Spreader rows? Viruliferous whiteflies? What is known about the susceptibility of the cassava varieties?
The experimental setup for such a virus study is not sufficient and warrants precise experiments. As such it is speculative to reflect on virus transmission.
3. The B. tabaci species was not determined, however a cassava adapted whitefly may have another settling behavior than a polyphagous visitor.
4. With all respect for cassava experiments, it is difficult to imagine how 250 adults can be traced in a choice experiment. The counts are very low although new B tabaci may have hatched. But is statistical significance among such low numbers relevant and in particular for transmission of a persistently transmitted virus.
5. Line 327-337 has to be clarified. Whiteflies prefer younger topmost leaves in general. What is meant with the initial leaf position and aversion of whiteflies towards first leaf position?
Minor
Line 50 B. tabaci the only vector of cassava geminiviruses, spread is predominantly by virus-infected cuttings
Line 52 cassava brown streak ipomovirus!
Line 67 . the main management for SLCMV is host plant resistance against viruses through CMD2 used throughout and introduced also to Thailand!
Line 90 … pioneered by (30) is a bit akward style
Author Response
Comments: The manuscript by Pimkornburee et al. reports studies on feeding behavior of B. tabaci on cassava using electrical penetration graphs and by measuring density and size of abaxial leaf trichomes using SEM. The findings of this study, that trichome size and density correlates with whitefly settling behavior can guide breeding strategies for whitefly resistance in cassava. A reduced number of whiteflies settling on cassava then has implications for transmission of viruses causing cassava mosaic disease, like SLCMV. All current Thai varieties are susceptible to SLCMV and deterring whiteflies from settling on cassava by breeding for a particular trichome size and density is very valid. Thus, the report is on an interesting subject, the manuscript is well written and the research is well described. However, there are several shortfalls the authors need to address before this article can be recommended for publication. They mostly concern the virus aspects of the work which need to be thoroughly revised and speculative statements removed, particularly when nothing is known about the experimental plants' susceptibility, virus titer etc (e.g. line 364 ….potentially contributing to the limited spread of SLCMV). |
Response: Thank you for pointing this out and providing outstanding suggestions that greatly improved the paper. Following the reviewer's suggestion, we have improved some weak points.
Major Comments 1: Virus detection by PCR is described in M&M, but there are no associated results/discussions. Response 1: The virus was detected to confirm the status of disease and non-disease cassava. Then, the non-disease cassavas were used for propagation and examined electropenetrography recording and whitefly B. tabaci settling experiment. We added the result of DNA gel electrophoresis of PCR amplification from cassava samples (Figure S1.) and described in Lines 420-422, page 13.
Comments 2: In 2.2. there is mention of severe virus infection in 30% in CMR-89 that is 6 plants of 20? a. Other varieties are also susceptible to SLCMV and but no infections in other varieties were reported? b. How can it be excluded that the CMR-89 plants were not infected from cuttings? Where did the virus come from? Spreader rows? Viruliferous whiteflies? What is known about the susceptibility of the cassava varieties? The experimental setup for such a virus study is not sufficient and warrant precise experiments. As such it is speculative to reflect on virus transmission. Response 2: Thank you for pointing this out. The SLCMD was severe in 30% of the CMR-89 cultivar, with 20 cassava plants, while the other varieties didn’t show symptoms. Although the other varieties show the status of whitefly number for settling under free choice in field conditions. However, SLCMV symptoms were only observed in CMR-89. This may be due to the latent period of disease in this variety, which we added to the discussion. We are confident that the CMR-89 plants are not infected with the cuttings because the plants were tested for infection before starting the experiment, and the experiment in the field was covered with a white net greenhouse, which is mentioned in line 473, page 14.
Comments 3: The B. tabaci species was not determined, however a cassava adapted whitefly may have another settling behavior than a polyphagous visitor. Response 3: In a preliminary study, we have previously surveyed and determined the whitefly B. tabaci species as cassava-living, feeding, and mating. Moreover, previous reports have studied them (Saokham et al., 2021).
|
Comments 4: With all respect for cassava experiments, it is difficult to imagine how 250 adults can be traced in a choice experiment. The counts are very low although new B tabaci may have hatched. But is statistical significance among such low numbers relevant and in particular for transmission of a persistently transmitted virus. Response 4: Thank you for pointing this out. From the results, we found that during 4-month-old plants, the CMR-89 showed moderate to severe mosaic symptoms. Whitefly-borne infections were less severe than infected cutting-borne infections. For clarity, we added a description and a literature review to the discussion lines 312 – 317, page 11.
|
Comments 5: Line 327-337 has to be clarified. Whiteflies prefer younger topmost leaves in general. What is meant with the initial leaf position and aversion of whiteflies towards first leaf position? Response 5: I apologize for writing this to mislead the reviewer. We mention that “This study exposed six cassava plants to field conditions infested with whiteflies. Interestingly, the initial leaf position of all six cassava cultivars showed lower whitefly infestation than the other leaf positions did.” Lines 342-344, page 12.
|
Minor Comments 6: Line 50 B. tabaci the only vector of cassava geminiviruses, spread is predominantly by virus-infected cuttings Line 52 cassava brown streak ipomovirus! Line 67 the main management for SLCMV is host plant resistance against viruses through CMD2 used throughout and introduced also to Thailand! Line 90 … pioneered by (30) is a bit akward style Response 6: Agree. We made corrections based on the reviewer's recommendations in the introduction part.
|

Reviewer 2 Report
Comments and Suggestions for Authors
In the reviewed MS a group of scientists from Thailand investigated stylet penetration behavior and settling of Bemisia tabaci, one of the most serious pests of Cassava, the important root crop. They tested six different cultivars of Cassava, differing in the densities of leaf trichomes and investigated the pattern of electric wavelength during feeding of Bemisia applying electrical penetration graph techniques. This is a very old method developed about 70 years ago for studying feeding process of various phytophagous insects. Previous authors concluded that the electric characteristics recorded during the feeding process of Bemisia depend on the properties of the protoplast of the most externals layer of plant cells [see references 30-40]. In the reviewed MS the authors found that it is also correlated with the density of trichomes investigated via SEM in this study. The authors also showed that the cultivars, which have higher density of trichomes, may be more resistant to the plant viruses, an important finding, previously demonstrated for several other “insect-plant” systems. The MS in general is well written and illustrated, although the MS would benefit if the authors inserted several high-quality photographs of the insect in the process of feeding and the photographs showing their equipment during the experiment. Captions for the Figures need some clarification, especially the abbreviations used in the Figures. Probably, the authors could give a list of the abbreviations in Material and Methods. The MS is a bit wordy. Additionally, some paragraphs are a bit hard to read. Perhaps, the authors could read the MS once again and decide which sentences they could simplify. The authors are also requested to explain better their idea to investigate virus negative and virus positive plant samples, which they detected using PCR. This goal was not specified in the Introduction and because of this, it is slightly surprising to see the paragraph on PCR in the section Material and Methods. Some information on the methods and different types of waveforms given in the Discussion could be transferred to Introduction. This would help a reader to understand the firther parts of the MS easier.
357 (Tjallingii et al. 2010) please, revise the reference format
Author Response
Comments 1: In the reviewed MS a group of scientists from Thailand investigated stylet penetration behavior and settling of Bemisia tabaci, one of the most serious pests of Cassava, the important root crop. They tested six different cultivars of Cassava, differing in the densities of leaf trichomes and investigated the pattern of electric wavelength during feeding of Bemisia applying electrical penetration graph techniques. This is a very old method developed about 70 years ago for studying feeding process of various phytophagous insects. Previous authors concluded that the electric characteristics recorded during the feeding process of Bemisia depend on the properties of the protoplast of the most externals layer of plant cells [see references 30-40]. In the reviewed MS the authors found that it is also correlated with the density of trichomes investigated via SEM in this study. The authors also showed that the cultivars, which have higher density of trichomes, may be more resistant to the plant viruses, an important finding, previously demonstrated for several other “insect-plant” systems. The MS in general is well written and illustrated, although the MS would benefit if the authors inserted several high-quality photographs of the insect in the process of feeding and the photographs showing their equipment during the experiment. Captions for the Figures need some clarification, especially the abbreviations used in the Figures. Probably, the authors could give a list of the abbreviations in Material and Methods. The MS is a bit wordy. Additionally, some paragraphs are a bit hard to read. Perhaps, the authors could read the MS once again and decide which sentences they could simplify. The authors are also requested to explain better their idea to investigate virus negative and virus positive plant samples, which they detected using PCR. This goal was not specified in the Introduction and because of this, it is slightly surprising to see the paragraph on PCR in the section Material and Methods. Some information on the methods and different types of waveforms given in the Discussion could be transferred to Introduction. This would help a reader to understand the firther parts of the MS easier.
357 (Tjallingii et al. 2010) please, revise the reference format
|
Response 1: Thank you for pointing this out. A reviewer complemented one another and provided outstanding suggestions that greatly improved the paper. |
For the virus-negative and virus-positive plant samples, the virus was detected to confirm the disease status and non-disease cassava by PCR, which is the process of disease status. Then, the non-disease cassavas were used for propagation, and the electropenetrography recording and whitefly B. tabaci settling experiment were examined. We added the DNA gel electrophoresis of PCR amplification from cassava samples in Figure S1 and described in Lines 420-422, page 13. We mention the caption for the EPG Figures in Figures 1 and 2. The waveform patterns of B. tabaci were categorized by amplitude, relative voltage level, R/emf origin, frequency, and the context of the waveform as non-probing (Np), stylet pathway (C), phloem salivation (E1), phloem ingestion (E2), intracellular puncture—potential drop (Pd), and xylem feeding (G). Moreover, the comments about the different types of waveforms given in the discussion could be transferred to the introduction. We agree; therefore, we added to the introduction. Line 95 – 99, page 2. |

Reviewer 3 Report
Comments and Suggestions for Authors
In this work, the authors used electrical penetration graph techniques to analyze the stylet penetration behavior of B. tabaci, in terms of settling and feeding in different Cassava cultivars. The results highlight the importance of the characteristics of the plant's trichomes in the settling preference of the whitefly. The work results are relevant, since aspects of the plant are identified that are crucial for developing varieties with resistance to the pest due to their morphological characteristics, which could also reduce CMD infection.
The literature review is extensive and relevant. The materials and methods are appropriate and adequately described. The results and discussion sections are developed in a detailed and complete manner. I therefore consider the work in its current form to be publishable.
Author Response
Comments 1: In this work, the authors used electrical penetration graph techniques to analyze the stylet penetration behavior of B. tabaci, in terms of settling and feeding in different Cassava cultivars. The results highlight the importance of the characteristics of the plant's trichomes in the settling preference of the whitefly. The work results are relevant, since aspects of the plant are identified that are crucial for developing varieties with resistance to the pest due to their morphological characteristics, which could also reduce CMD infection.
The literature review is extensive and relevant. The materials and methods are appropriate and adequately described. The results and discussion sections are developed in a detailed and complete manner. I therefore consider the work in its current form to be publishable. |
Response 1: Thank you
|
